# Medical Advocacy among Latina Women Diagnosed with Breast Cancer

**DOI:** 10.3390/ijerph21040495

**Published:** 2024-04-18

**Authors:** Paola Torres, Judith Guitelman, Araceli Lucio, Christine Rini, Yamilé Molina

**Affiliations:** 1University of Illinois Cancer Center, University of Illinois, Chicago, IL 60612, USA; ptorres4@uic.edu; 2Asociación Latina de Asistencia y Prevención del Cáncer de Mama (ALAS-WINGS), Chicago, IL 60657, USA; jguitelman@alas-wings.org; 3The Resurrection Project, Chicago 60608, IL, USA; 4Department of Medical Social Sciences, Feinberg School of Medicine, Northwestern University, Chicago, IL 60611, USA; 5Division of Community Health Sciences, School of Public Health, University of Illinois Chicago, Chicago, IL 60612, USA; 6Hospital & Health Sciences Systems Mile Square Health Center, University of Illinois, Chicago, IL 60612, USA

**Keywords:** breast cancer, health disparities, Latinas, medical advocacy, survivorship, community health research

## Abstract

Medical advocacy has continued to significantly impact quality of life and survivorship outcomes among Latina breast cancer survivors in the United States. However, little is known about the unique experiences of Latina survivors, including the perceived value, process, and context in which they practice medical advocacy. To help address this gap, we conducted a qualitative, secondary analysis of semi-structured focus groups with 18 Latina breast cancer survivors from Chicago, Illinois. Eligible women had to self-identify as (1) female, (2) Latina, (3) 18 years or older, and (4) having a breast cancer diagnosis 5 years ago or more. In total, 61% of participants were 50–59 years old, 83% were born in Mexico, and 100% spoke Spanish. The three emergent themes from the focus groups were (1) the cultural need for Latina advocates and support groups; (2) the process and experiences of becoming a community advocate within Latine culture; and (3) the cultural contexts for advocacy by Latina breast cancer survivors. Latina survivor advocates share strengths of receiving ongoing health education, peer support, and access to resources when being linked to a support group furthering their exposure to role models, increasing their awareness of opportunities in medical advocacy, and providing an entry to participate in medical advocacy.

## 1. Introduction

Breast cancer is the leading cause of cancer death for Latina women in the United States [1,2]. Although Latinas have a 30% lower incidence rate of breast cancer than their non-Latina white (NLW) counterparts, Latinas are more likely to be diagnosed at more advanced stages, lead a lower quality of life as a breast cancer survivor, and are about 30% more likely to die from their breast cancer than their non-Latina white counterparts [3]. In response, numerous interventions across multiple decades have sought to improve early breast cancer detection, quality of life, and survivorship outcomes among Latinas [4,5,6]. Latina breast cancer survivors’ medical advocacy has been significant for this body of work. Here, we define medical advocacy as “focused actions and work of supporters from various walks of life, including cancer survivors and their loved ones, civil society organizations, clinicians, and researchers” [7,8]. Many of the aforementioned interventions and programs have leveraged Latina advocates, through sharing their testimonies, serving as community health workers to promote mammography, and providing peer support to others diagnosed with breast cancer [9]. Thus, Latina women who are recipients of these services have the potential to become advocates themselves, furthering the multilevel impacts of these programs.

The value and impact of medical advocacy to achieve breast cancer equity has been increasingly studied [10,11,12] including among African American survivors and other marginalized populations [7,12,13,14,15,16,17,18,19]. Past research has identified potentially universal experiences across racial and ethnic groups (e.g., reciprocity, or the value of “giving back”) and culturally specific experiences with medical advocacy (e.g., navigating community mistrust, the need for minority breast health spaces/support groups). Yet we are aware of no studies to date that have explored the unique experiences of Latina breast cancer survivors, including the perceived value, process, and context in which they practice medical advocacy. This represents an important dearth in the literature, especially given the unique stressors and intersectional identities many Latinas face as immigrants, children of immigrants, and individuals with strong bi-national family ties and roots [20,21]. Addressing these gaps in the literature can contribute to current health equity initiatives, including improving how we recruit Latina survivors in Latina-based community breast health programs; how we support them as advocates; and how we measure their reach and impact among Latino communities.

To begin to address this knowledge gap, the current study examines the perceived value, process, and context in which Latina breast cancer survivors engage in medical advocacy. We focus our efforts on community/interpersonal advocacy [22], which we define here as the efforts of individuals to optimize their communities’ access, uptake, and experiences with breast healthcare.

## 2. Materials and Methods

### 2.1. Setting

The current study was conducted using a virtual communications platform among breast cancer survivors within Chicago, IL, USA.

### 2.2. Design

We conducted a qualitative, secondary analysis of semi-structured focus groups with 18 Latina breast cancer survivors from Chicago, IL, USA.

### 2.3. Procedures

The current secondary analysis was based on a larger qualitative study, which focused on soliciting patient input for adaptation of existing peer-based survivorship interventions. The project included community (Guitelman, Lucio) and academic co-investigators (Molina, Rini, Torres). All methods and materials were approved by the University of Illinois at Chicago’s Institutional Review Board (STUDY2021-0584).

We conducted focus groups between June and August 2021, using a multi-frame convenience sampling strategy to recruit participants via community investigators’ virtual events, newsletters, and websites. To be eligible, participants had to self-identify as (1) female, (2) Latina, (3) 18 years or older, and (4) having a breast cancer diagnosis 5 years ago or more.

Interested Latina breast cancer survivors contacted staff by phone to confirm eligibility and were scheduled to participate in virtual focus groups. As described below, most participants indicated a preference for Spanish-based focus groups. Accordingly, we conducted two one-hour Spanish-based focus groups with 8–10 individuals per group. Prior to the start of focus groups, the study team obtained verbal consent and administered an anonymous poll to obtain the demographic (e.g., age, country of birth, insurance status, marital status) and cancer–related factors (e.g., years since diagnosis, treatment history). Study staff also gauged participant’s comfort levels in using the virtual communications platform and ensured setting ground rules before starting any discussions (i.e., staying on mute while others were talking, remaining active and on-camera). Subsequently, bilingual, bicultural staff led semi-structured discussions. Two example questions that the participants were asked are, “How have [survivorship] programs [organizations] helped in terms of having a space to reflect on your experiences with breast cancer?” and “How have you used your testimonials and personal experiences when talking about mammography to others?” Focus groups lasted approximately 90 min. Participants received $50 for their time and effort.

### 2.4. Qualitative Data Analysis

Focus groups were audio-recorded, transcribed verbatim, and uploaded into qualitative analytic software. A team of two coders (PT, YM) led a content analysis using both deductive (theory-driven) and inductive (themes emerging from iterative analysis) analytic approaches [23]. Our initial coding scheme identified unique aspects of medical advocacy among Latina survivors using deductive codes drawn from theoretical frameworks on medical advocacy [13,22]. We then identified new inductive themes from raw focus group data. Each transcript was read independently and then discussed in group settings to ensure consistent interpretation and foster inter-coder reliability. Disagreements were resolved by reviewing transcripts and discussing perspectives until consensus was reached. Coders then grouped similar concepts into categories that illustrated the identified themes. We did not quantify information regarding our qualitative data, given that this analysis and the larger qualitative study were not designed to enumerate associations [24]. Providing percentages would have led to misleading information regarding the concrete frequency and magnitude of themes [25,26].

## 3. Results

### 3.1. Study Sample Characteristics

Sample characteristics are depicted in Table 1. Overall, 61% of our participants were 50–59 years old, 83% were born in Mexico, 100% spoke Spanish, 83% were married, and about two-thirds of participants had no insurance or had public insurance (e.g., Medicaid, Medicare). Over half of the sample were diagnosed ≥ 11 years ago. All survivors had undergone surgery, with 61% also receiving chemotherapy and 67% also receiving radiation.

### 3.2. Emergent Themes

We grouped emergent themes into three categories concerning the unique aspects and manifestations of advocacy among Latina breast cancer survivors: (1) the cultural need for Latina advocates and support groups; (2) the process and experiences becoming a community advocate within Latine culture; and (3) the cultural contexts for advocacy by Latina breast cancer survivors.

#### 3.2.1. The Cultural Need for Latina Advocates and Support Groups

Access to Latina advocates and support groups was perceived to be particularly important in the context of migrant loneliness and language barriers. One survivor noted, “I did not share it with my family in Mexico…you don’t want to see them suffer, because they know your situation…you don’t want to hurt them, by letting them know what you are going through”. Another survivor similarly reflected, “I think that’s what stops us, as a Hispanic culture. We don’t want to make our family suffer or make them suffer with us”. Others emphasized cultural stigmas, which impeded their ability to engage their social networks, as exemplified by this respondent:

I think the fear, the fear of sharing it [news of breast cancer diagnosis], is the fear of being judged…I don’t know what the culture of others is like, let’s say of the Anglo-Saxons here. But we do [judge], because they point at you, because you see it… many people believe, many people believe that it sticks. Even nowadays, there are still people who believe that cancer sticks.

Survivors emphasized the need and value of Latina-based groups, given this unique geographic and cultural isolation, which was often compounded by language barriers faced in cancer survivorship services. 

It is very important for us Latinas that we can live together and we can express all our feelings, because the moment we are told that we have cancer it seems that the world is coming down on us and the first thing we think of is death. In this country, I am alone, with my children…and sometimes there is no one to talk to or one does not have that type of relationship with people. The support group is very important…because there, we have many tools, we no longer feel alone, [and] no one can understand us more than the people who went through the same process. I think that these [support] groups should be extended, that there should be more in Spanish…I notice in the web pages that there are groups that have many events, but almost all of them are in English. They do not have in Spanish and I think that what we need is to have more, more events in Spanish or more groups.

#### 3.2.2. The Process and Experience of Becoming a Community Advocate within Latine Culture

The process of becoming a community advocate, similar to past research, often began in support groups and through meeting other survivors involved in advocacy work. A survivor identified, “I think [survivor testimonies] are important, because they see that you are Latina and that you are a survivor. So those are already two factors that they take into account—if she could do it, I can do it too”. In line, past experiences with facing cultural barriers and myths (e.g., “They start giving you a thousand remedies”) motivated participation in community advocacy as a survivor. For example, one respondent noted, “If we stay like this [not talking about cancer], then our community will always be ignorant. So that is what we are called to do…we are cancer survivors…this is our calling”. Others were encouraged by their experiences after treatment, as one survivor shared:

When I had the surgery I was completely lost. I had [the] surgery, I went home, they gave me very little information [in Spanish]. They gave me a very big book, a lot of information, but information that was in English which I could not read and with the stress of the surgery, of the cancer, of everything, I did not have that part of reading what I was going to do after the surgery. So, I think it is perfect, maybe if my testimony can help other people or also give them more information in their language, well, I say it would be perfect, because then they will be more informed, because I was very uninformed, very lost.

Survivors shared the cultural barriers and myths they often encountered after beginning to participate in community advocacy. One survivor declared, “I have really enjoyed participating and bringing information to other women, because there has been a lot of taboo among us Latinas…we encountered many women who did not even want you to talk to them about mammography”. Common linkages between cancer and death were brought up by survivors, one saying “Maybe we don’t want to talk about cancer because it is like talking about death; but we shouldn’t do it. We should inform ourselves a little bit more”. Survivors also mentioned overcoming discomfort due to unfamiliar situations, “I was embarrassed to talk about it in front of men, because it was the first time, yes I was embarrassed to talk about it in front of men, but they also have to know. They should also know and learn from other people’s experiences”. Others shared their growth, given their increasing need to address barriers to access and other structural barriers. One respondent highlighted:

We have to keep going forward helping other people to let them know that they are not alone, whether they have insurance or not. [This is] precisely the information to look for [to help] people–to look for organizations that help for free mammograms, hospitals that also provide them. It is all a matter of informing and searching.

#### 3.2.3. The Cultural Contexts for Advocacy by Latina Breast Cancer Survivors

Our participants highlighted Latinas’ unique opportunities to participate in medical advocacy. For example, several survivors participated in breast health promotion in the local Mexican Consulate. This context enabled them to engage Latinas who may not participate in traditional venues for cancer education (e.g., health fairs, primary care doctors), as reflected by these two survivors.

At the consulate, one of them ran away…when I said I was a cancer survivor, she got out of line and went to the stairs of the consulate. She didn’t want to hear anything about cancer. I had the microphone, anyway, [so] she heard. It helps you a lot, it gives you the power, to be able to share your experience to also empower these women to speak up for themselves, to advocate for their health. Because they keep quiet, but in reality there are times when they have something. They have a concern and when you speak, you can see that concern in their face. And sometimes they ask you: ‘Hey, how did you do it? How did you get a mammogram?’ In other words, they start asking questions and that’s where you feel good, because you can help and it helps a lot inside you and at the same time it strengthens you to be able to help. The fact that we got cancer changed our lives.

We are going to fight to help people who are negative. Because there are many people who do not want to–it happened to me at the consulate. I met a lady and she told me: ‘No, I do not believe in mammograms, pap smears, I do not go to the doctor because I do not like to be seen by the doctor’. Then her husband came up to me and said, ‘Can you make an appointment for my wife?’ he said. “No, I don’t make appointments, I just filled out the sheet for her to fill out all the information and they [the clinic] will talk to her to make an appointment’…I gave it to him and they [the clinic] called the lady and she had her [mammogram] done. Now she says…to her daughters ‘Get the [mammogram] done!

As immigrants, Latina breast cancer survivors further shared unique opportunities for global breast health promotion through their bi-national networks. One survivor emphasized, “I even called my family in Mexico, I called everybody and I told them: ‘Check yourselves, look, I was diagnosed’”.

## 4. Discussion

Latina breast cancer survivors experience a wide set of multilevel system barriers upon receiving a breast cancer diagnosis. Simultaneously, Latina survivor advocates serve as a cornerstone to community and multi-sectoral efforts to address non-Latina white-Latina disparities across the breast cancer outcome and care continuum [5,6,7,8]. Thus, interventions that promote medical advocacy among Latina women diagnosed with breast cancer may have benefits for women’s quality of life, other patients, and their social networks [7]. Our study provides value toward this goal by offering testable hypothesis for future testing and insights into the perceived value, processes, and natural contexts in which Latina breast cancer survivors engage in medical advocacy. Such work offers important insights for future engagement and retention of Latina survivor advocates in the breast health equity workforce.

Survivors in our study emphasized the need and cultural value of Latina breast cancer advocates and support groups. The value of Latina breast cancer survivorship resources was framed in the context of unique stressors, such as geographic isolation for migrant women, language barriers to access available survivorship support services, and cultural isolation due to stigma and misconceptions about breast cancer. Further, survivors’ limited disclosure and access to support for their diagnoses may reflect unique cultural values, such as marianismo, wherein Latinas are socialized to be quiet and not speak about their opinions [27]. Such cultural values likely underlie Latinas’ decisions to not disclose their diagnoses and limit their access to support during a critical, traumatic time. Past research in Latina breast cancer survivorship has highlighted these distinct stressors and the value of Latina-based resources in the context of care [28,29]. Our work adds to this important literature by framing the type of supportive environments that support Latina breast cancer survivors and enable them to become change agents themselves, despite cultural limitations.

Our research suggests that supportive environments for nourishing medical advocacy include linking Latina breast cancer survivors with Latina community health workers, other breast cancer survivors, and support groups. These findings align with previous research highlighting unintended health-protective consequences among African American breast cancer survivors whose positive experiences in supportive environments instilled a desire to advocate for others [13,15,22]. Through these programs, breast cancer survivors can receive highly valued ongoing education on their own health and diagnoses; peer support with dismantling social stigma surrounding cancer; and access to resources (e.g., financial support, etc.) [9,30,31]. Although further research is needed to study the impact advocates or support groups have on treatment decision-making, most participants disclosed having a better understanding of their diagnosis after participating in support groups and community organizations that motivated them to share their experiences either informally (with co-workers, friends, and family on a day-to-day basis) or more formally (through affiliations with established organizations), showing the wide range of reach survivors have with individuals in the community.

Support groups can further offer exposure to role models, awareness of opportunities in medical advocacy, and an entry to participate in medical advocacy. Survivors also reported being driven to serve as advocates after being a target of many Latine cultural misconceptions about breast cancer (e.g., infectious diseases) and associated stigma. In attempts to break cultural cycles, survivors were motivated to be advocates, and they were particularly well-equipped to handle misconceptions and stigma, as new community leaders with personal, direct experience with breast cancer. Altogether, these findings highlight opportunities for recruiting, training, and elevating Latina advocates from culturally congruent support groups. Simultaneously, our findings suggest that new Latina advocates may benefit, in particular, from continuous support through these groups, through building shared norms around advocacy as well as survivorship.

Dismantling the negative connotations associated with breast cancer is a multifaceted endeavor that requires collective action and advocacy. Latina breast cancer survivors played a crucial role in fostering a culture of empathy and solidarity within their communities. They were able to create spaces for open dialogue, support, and empowerment, as well as challenge stigma and foster greater understanding of breast cancer. Through personal experiences and storytelling, survivors were able to confront cultural taboos and stigmas surrounding the word “cancer”, paving the road for greater awareness and understanding among their friends, family, and community members. As exemplified in our study, medical advocacy emerged as a powerful tool in the process of diminishing the negative impact of referencing cancer. In leveraging their advocacy efforts, survivors not only educate their community about the realities of cancer, but also advocate for improved access to healthcare services, culturally competent care, and survivorship support.

Our study offers intriguing findings about the reach of Latina breast cancer survivors within local and global settings. A significant body of research has highlighted the challenges of bi-national networks, especially in the context of social isolation [32,33]. Our findings, however, highlight the strength of bi-national networks, in that Latina survivor advocates may be able to reach family and friends in their countries of origin, wherein there may be limited community health promotion and resources for breast healthcare. Further, our findings suggest that Latina breast cancer survivors may have unique, understudied, niches for breast health promotion through their identities and connections with larger Latina immigrant communities. Future studies may benefit from leveraging Latina cancer survivor advocates in the context of consulates and other venues for immigrant communities.

### Limitations

The current project has several limitations. Some features of this study may limit generalizability given that this study focused solely on Latina women residing in an urban city in the United States. Furthermore, 83% of the participants in the study identified as Mexican which is representative of those who identify as Latine in Chicago, IL wherein those with Mexican descent make up 73.7% of the population [34]. Nonetheless, themes identified may exist for Latinas in other cultural and geographical locations. Multiethnic studies samples should also be conducted to explore experiences for other racial/ethnic groups. Participants in the study had some previous involvement in support groups, which may not be representative of the experiences of diagnosed Latina women who may not have been linked to support groups in the past. Interviews were conducted through a cloud-based video conferencing platform, limiting participation of individuals who may not have internet or computer access and/or lack proficiency with technology. Although incentivizing participants in research studies has raised ethical concerns, participants in this study were compensated $50, which is within the standard range of compensation offered in qualitative research [35]. However, compensation may have influenced participant engagement. The purpose of the larger qualitative study was to obtain feedback about how to adapt existing intervention materials for future trials, which may have restricted the types of themes that could emerge from this secondary analysis. With a sample size of 18, the current project highlights the importance of future, quantitative studies with large sample sizes to test the hypotheses generated from this study. Finally, some of the participants had been diagnosed over 5 years ago, which may contribute to recall bias.

## 5. Conclusions

Latinas continue to experience barriers to cancer care post-diagnosis; however, breast cancer support groups provide tools to promote resiliency and survivorship among Latina breast cancer survivors. Findings inform the process in which Latina breast cancer survivors overcome cultural barriers after a cancer diagnosis whilst becoming community advocates and agents of change within their respective communities and cultures. From receiving health education, peer support, and access to resources, Latina breast cancer survivors feel better equipped to share their experiences formally and informally with family, friends, or through affiliations with community organizations. By amplifying their voice and advocating for others, survivors play a pivotal role in shaping the narrative surrounding breast cancer within their communities. Although the reach of Latina breast cancer survivors in this study includes local and global settings, future studies may benefit from exploring the impact Latina cancer advocates have on locations catering to immigrant communities such as consulates. Moreover, future quantitative studies with a larger sample size could validate the themes identified in this study, offering a testable hypothesis for further investigation. By continuing to explore the experiences and contributions of Latina breast cancer survivors and their role in medical advocacy, stakeholders can work collaboratively to advance health equity and promote a more inclusive approach to breast cancer care.

## Figures and Tables

**Table 1 ijerph-21-00495-t001:** Study sample demographic and cancer-related factors.

Variables	Categories	Sample (n = 18)
Demographic factors		
Age (years)	50–59	11 (61%)
	60–69	5 (28%)
	70–79	2 (11%)
Country of Birth	Mexico	15 (83%)
	United States	1 (5%)
	Honduras	1 (6%)
	Guatemala	1 (6%)
Language(s) Spoken	English	5 (28%)
	Spanish	18 (100%)
Marital Status	Married	15 (83%)
	Unmarried	3 (17%)
Insurance	Private	6 (33%)
	Public	7 (39%)
	None	5 (28%)
Education Level	Less than 5th grade	1 (6%)
	5th–8th grade	2 (11%)
	9th–12th grade	5 (28%)
	GED	2 (11%)
	Some College	8 (44%)
Cancer-Related Factors		
Years Since Diagnosed	5–6 years	4 (22%)
	7–8 years	2 (11%)
	9–10 years	2 (11%)
	More than 11 years	10 (56%)
Treatment Received	Chemotherapy	11 (61%)
	Surgery	18 (100%)
	Radiation	12 (67%)

## Data Availability

Interested parties are encouraged to contact Molina (ymolin2@uic.edu) for access to de-identified qualitative data.

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
