# Peer review of "Medical Advocacy among Latina Women Diagnosed with Breast Cancer"

_ijerph, 2024, doi:10.3390/ijerph21040495_

Round 1
Reviewer 1 Report
Comments and Suggestions for Authors
Dear Authors,
Great job conducting this study that examines the perceived values, process, and context in which Latina Breast Cancer Survivors engage in medical advocacy.
Overall, the manuscript was well written, with consistency of tone and flow. The research method utilized is appropriate and the presentation of findings is clear.
Just a few comments/observations and suggestions for your consideration.
Background: This section highlighted the breast cancer as a leading cause of death among Latina women, and delved into the ideal of survivorship, including the role of advocates (who are also survivors) and the ripple effect this informal service (of care) has on the population. The review of literature is appropriate. Great job identifying the knowledge gaps.
- Is there data on the prevalence of breast cancer and Breast Ca survivors? Consider including these statistics somewhere around Lines 30 – 32.
- Consider moving the last sentence (lines 65 – 67) to the Materials and Methods section.
Methods:
As suggested above, consider stating the study design here.
The population of interest and eligibility criteria are appropriate.
The ethical consideration, including the acquisition of verbal consent is appropriate, such as is common in virtual focus groups.
- Consider including a sentence about the length of time it took to complete each focus group discussion.
The content of the data analysis section is appropriate.
Result: The themes are appropriate.
Discussion:
- Consider citing how support groups have been instrumental to dealing with stigmatization in the Latina population and other cultural groups.
- Would also be interesting to see how medical advocacy may have thrived among other ethnocultural groups.
- Re: table 1 noted that 100% of participants had surgery. Consider including information on how advocates contribute to the acceptance of treatment options. Please remember to thorough reference the literature on the impact of ethnoculturally-inclined medical advocates.
Author Response
Dear Reviewer,
Thank you for your helpful comments and suggestions. Please see document attached for my response and actions taken. I look for forward to your response.

Reviewer 2 Report
Comments and Suggestions for Authors
General comments:
In this articles authors investigated medical advocacy for breast cancer survivors among Latina women. This study is based on data originating from semi structure focus group of 18 women diagnosed with breast cancer and fitting selection criteria. Authors identified and classified the result in three themes from the focus group. The importance and need to Latina advocates and support groups due to cultural and geographical isolation. The process and experience of becoming a community advocate in which advocates have to overcome cultural myths and barriers. Lastly authors mentioned the contexts for advocacy by Latina breast cancer survivors.
Overall, the manuscript is well written and clear to follow. Giving important hindsight into breast cancer advocacy among Latina. The methodological reasoning is clearly disclosed and described in the manuscript. The discussion is the real strength of the manuscript and most of the comments raised while reading the manuscript were stated and addressed in this section as well as in the limitation section.
Acceptance with minor revision is recommended.
Minor comment:
Comment #1 While this information is implicit, the country in which the study is taking place should be clearly mentioned in abstract and introduction.
Comment #2 Following previous comment, could the authors comment on the potential geographic limitation of the study?
Comment #2 Furthermore could the authors elaborate on the cultural limitation that might represents the skew in the cohort (83% born in Mexico)?
Author Response

(The authors gave the same response as above.)

Reviewer 3 Report
Comments and Suggestions for Authors
While your exploration of the unique experiences of Latina breast cancer survivors, particularly in the realm of medical advocacy, is intriguing, I've identified two significant shortcomings in your study. Firstly, despite the fact that breast cancer is among the most frequent cancers in women, your study population is of only 18 women, which I consider it to be insufficient to draw any valid conclusion. Secondly you mentioned that the participant received 50$ a practice that raises ethical concerns.
Nonetheless the topic is an interesting one but you need a larger study population to support your results.
Author Response

(The authors gave the same response as above.)

Round 2
Reviewer 3 Report
Comments and Suggestions for Authors
My main concern, the small study population of only 18 women, remains and it is a matter that can only be addressed by including more women in the study. In the end I think it is up to the Editor to decide if the article presents relevant and valuable data given this limitation and if it should be published.
Author Response
Dear Reviewer and Editor,
Thank you for your time and feedback on our manuscript. Our response letter is attached below. If you are to have additional questions or feedback we would be happy to discuss.
Kind regards,
Paola
